# Tight Bounds for Influence in Diffusion Networks and Application to Bond Percolation and Epidemiology

**Rémi Lemonnier**[1,2]       **Kevin Scaman**[1]       **Nicolas Vayatis**[1]

[1]CMLA – ENS Cachan, CNRS, France, [2]1000mercis, Paris, France

`{lemonnier, scaman, vayatis}@cmla.ens-cachan.fr`

## Abstract

In this paper, we derive theoretical bounds for the long-term influence of a node in an Independent Cascade Model (ICM). We relate these bounds to the spectral radius of a particular matrix and show that the behavior is sub-critical when this spectral radius is lower than $1$. More specifically, we point out that, in general networks, the sub-critical regime behaves in $O(\sqrt{n})$ where $n$ is the size of the network, and that this upper bound is met for star-shaped networks. We apply our results to epidemiology and percolation on arbitrary networks, and derive a bound for the critical value beyond which a giant connected component arises. Finally, we show empirically the tightness of our bounds for a large family of networks.

## 1 Introduction

The emergence of social graphs of the World Wide Web has had a considerable effect on propagation of ideas or information. For advertisers, these new diffusion networks have become a favored vector for *viral marketing* operations, that consist of advertisements that people are likely to share by themselves with their social circle, thus creating a propagation dynamics somewhat similar to the spreading of a virus in epidemiology ([1]). Of particular interest is the problem of *influence maximization*, which consists of selecting the top-k nodes of the network to infect at time $t = 0$ in order to maximize in expectation the final number of infected nodes at the end of the epidemic. This problem was first formulated by Domingues and Richardson in [2] and later expressed in [3] as an NP-hard discrete optimization problem under the Independent Cascade (IC) framework, a widely-used probabilistic model for information propagation.

From an algorithmic point of view, influence maximization has been fairly well studied. Assuming the transmission probability of all edges are known, Kempe, Kleinberg and Tardos ([3]) derived a greedy algorithm based on Monte-Carlo simulations that was shown to approximate the optimal solution up to a factor $1 - \frac{1}{e}$, building on classical results of optimization theory. Since then, various techniques were proposed in order to significantly improve the scalability of this algorithm ([4, 5, 6, 7]), and also to provide an estimate of the transmission probabilities from real data ([8, 9]). Recently, a series of papers ([10, 11, 12]) introduced *continuous-time* diffusion networks in which infection spreads during a time period $T$ at varying rates across the different edges. While these models provide a more accurate representation of real-world networks for finite $T$, they are equivalent to the IC model when $T \to \infty$. In this paper, will focus on such long-term behavior of the contagion.

From a theoretical point of view, little is known about the influence maximization problem under the IC model framework. The most celebrated result established by Newman ([13]) proves the equivalence between bond percolation and the *Susceptible-Infected-Removed* (SIR) model in epidemiology ([14]) that can be identified to a special case of IC model where transmission probability are equal amongst all infectious edges.

In this paper, we propose new bounds on the influence of any set of nodes. Moreover, we prove the existence of an *epidemic threshold* for a key quantity defined by the spectral radius of a given *hazard*

*matrix*. Under this threshold, the influence of *any* given set of nodes in a network of size $n$ will be $O(\sqrt{n})$, while the influence of a randomly chosen set of nodes will be $O(1)$. We provide empirical evidence that these bounds are sharp for a family of graphs and sets of initial influencers and can therefore be used as what is to our knowledge the first closed-form formulas for influence estimation. We show that these results generalize bounds obtained on the SIR model by Draief, Ganesh and Massoulié ([15]) and are closely related to recent results on percolation on finite inhomogeneous random graphs ([16]).

The rest of the paper is organized as follows. In Sec. 2, we recall the definition of Information Cascades Model and introduce useful notations. In Sec. 3, we derive theoretical bounds for the influence. In Sec. 4, we show that our results also apply to the fields of percolation and epidemiology and generalize existing results in these fields. In Sec. 5, we illustrate our results by applying them on simple networks and retrieving well-known results. In Sec. 6, we perform experiments in order to show that our bounds are sharp for a family of graphs and sets of initial nodes.

## 2 Information Cascades Model

### 2.1 Influence in random networks and infection dynamics

Let $\mathcal{G} = (\mathcal{V}, \mathcal{E})$ be a directed network of $n$ nodes and $A \subset \mathcal{V}$ be a set of $n_0$ nodes that are initially *contagious* (e.g. aware of a piece of information, infected by a disease or adopting a product). In the sequel, we will refer to $A$ as the *influencers*. The behavior of the cascade is modeled using a probabilistic framework. The influencer nodes spread the contagion through the network by means of transmission through the edges of the network. More specifically, each contagious node can infect its neighbors with a certain probability. The *influence* of $A$, denoted as $\sigma(A)$, is the expected number of nodes reached by the contagion originating from $A$, i.e.

$$\sigma(A) = \sum_{v \in \mathcal{V}} \mathbb{P}(v \text{ is infected by the contagion } | A). \tag{1}$$

We consider three infection dynamics that we will show in the next section to be equivalent regarding the total number of infected nodes at the end of the epidemic.

**Discrete-Time Information Cascades [$DTIC(\mathcal{P})$]** At time $t = 0$, only the influencers are infected. Given a matrix $\mathcal{P} = (p_{ij})_{ij} \in [0,1]^{n \times n}$, each node $i$ that receives the contagion at time $t$ may transmit it at time $t + 1$ along its outgoing edge $(i, j) \in \mathcal{E}$ with probability $p_{ij}$. Node $i$ cannot make any attempt to infect its neighbors in subsequent rounds. The process terminates when no more infections are possible.

**Continuous-Time Information Cascades [$CTIC(\mathcal{F}, T)$]** At time $t = 0$, only the influencers are infected. Given a matrix $\mathcal{F} = (f_{ij})_{ij}$ of non-negative integrable functions, each node $i$ that receives the contagion at time $t$ may transmit it at time $s > t$ along its outgoing edge $(i, j) \in \mathcal{E}$ with stochastic rate of occurrence $f_{ij}(s - t)$. The process terminates at a given deterministic time $T > 0$. This model is much richer than Discrete-time IC, but we will focus here on its behavior when $T = \infty$.

**Random Networks [$RN(\mathcal{P})$]** Given a matrix $\mathcal{P} = (p_{ij})_{ij} \in [0,1]^{n \times n}$, each edge $(i, j) \in \mathcal{E}$ is removed independently of the others with probability $1 - p_{ij}$. A node $i \in \mathcal{V}$ is said to be *infected* if $i$ is linked to at least one element of $A$ in the spanning subgraph $\mathcal{G}' = (\mathcal{V}, \mathcal{E}')$ where $\mathcal{E}' \subset \mathcal{E}$ is the set of non-removed edges.

For any $v \in \mathcal{V}$, we will designate by *influence of* $v$ the influence of the set containing only $v$, i.e. $\sigma(\{v\})$. We will show in Section 4.2 that, if $\mathcal{P}$ is symmetric and $\mathcal{G}$ undirected, these three infection processes are equivalent to *bond percolation* and the influence of a node $v$ is also equal to the expected size of the *connected component* containing $v$ in $\mathcal{G}'$. This will make our results applicable to percolation in arbitrary networks. Following the percolation literature, we will denote as *sub-critical* a cascade whose influence is not proportional to the size of the network $n$.

## 2.2 The hazard matrix

In order to linearize the influence problem and derive upper bounds, we introduce the concept of *hazard matrix*, which describes the behavior of the information cascade. As we will see in the following, in the case of Continuous-time Information Cascades, this matrix gives, for each edge of the network, the integral of the instantaneous rate of transmission (known as hazard function). The spectral radius of this matrix will play a key role in the influence of the cascade.

**Definition.** For a given graph $\mathcal{G} = (\mathcal{V}, \mathcal{E})$ and edge transmission probabilities $p_{ij}$, let $\mathcal{H}$ be the $n \times n$ matrix, denoted as the *hazard matrix*, whose coefficients are

$$\mathcal{H}_{ij} = \begin{cases} -\ln(1 - p_{ij}) & \text{if } (i, j) \in \mathcal{E} \\ 0 & \text{otherwise} \end{cases} . \tag{2}$$

Next lemma shows the equivalence between the three definitions of the previous section.

**Lemma 1.** *For a given graph* $\mathcal{G} = (\mathcal{V}, \mathcal{E})$, *set of influencers* $A$, *and transmission probabilities matrix* $\mathcal{P}$, *the distribution of the set of infected nodes is equal under the infection dynamics* $DTIC(\mathcal{P}), CTIC(\mathcal{F}, \infty)$ *and* $RN(\mathcal{P})$, *provided that for any* $(i, j) \in \mathcal{E}$, $\int_0^\infty f_{ij}(t)dt = \mathcal{H}_{ij}$.

**Definition.** For a given set of influencers $A \subset \mathcal{V}$, we will denote as $\mathcal{H}(A)$ the hazard matrix except for zeros along the columns whose indices are in $A$:

$$\mathcal{H}(A)_{ij} = \mathbb{1}_{\{j \notin A\}} \mathcal{H}_{ij}. \tag{3}$$

We recall that for any square matrix $M$, its spectral radius $\rho(M)$ is defined by $\rho(M) = \max_i(|\lambda_i|)$ where $\lambda_1, ..., \lambda_n$ are the (possibly repeated) eigenvalues of matrix $M$. We will also use that, when $M$ is a real square matrix with positive entries, $\rho(\frac{M+M^\top}{2}) = \sup_X \frac{X^\top M X}{X^\top X}$.

**Remark.** When the $p_{ij}$ are small, the hazard matrix is very close to the transmission matrix $\mathcal{P}$. This implies that, for low $p_{ij}$ values, the spectral radius of $\mathcal{H}$ will be very close to that of $\mathcal{P}$. More specifically, a simple calculation holds

$$\rho(\mathcal{P}) \leq \rho(\mathcal{H}) \leq \frac{-\ln(1 - \|\mathcal{P}\|_\infty)}{\|\mathcal{P}\|_\infty} \rho(\mathcal{P}), \tag{4}$$

where $\|\mathcal{P}\|_\infty = \max_{i,j} p_{ij}$. The relatively slow increase of $\frac{-\ln(1-x)}{x}$ for $x \to 1^-$ implies that the behavior of $\rho(\mathcal{P})$ and $\rho(\mathcal{H})$ will be of the same order of magnitude even for high (but lower than 1) values of $\|\mathcal{P}\|_\infty$.

## 3 Upper bounds for the influence of a set of nodes

Given $A \subset \mathcal{V}$ the set of influencer nodes and $|A| = n_0 < n$, we derive here two upper bounds for the influence of $A$. The first bound (Proposition 1) applies to any set of influencers $A$ such that $|A| = n_0$. Intuitively, this result correspond to a best-case scenario (or a worst-case scenario, depending on the viewpoint), since we can target any set of nodes so as to maximize the resulting contagion.

**Proposition 1.** *Define* $\rho_c(A) = \rho(\frac{\mathcal{H}(A) + \mathcal{H}(A)^\top}{2})$. *Then, for any $A$ such that* $|A| = n_0 < n$, *denoting by $\sigma(A)$ the expected number of nodes reached by the cascade starting from $A$:*

$$\sigma(A) \leq n_0 + \gamma_1(n - n_0), \tag{5}$$

*where $\gamma_1$ is the smallest solution in $[0, 1]$ of the following equation:*

$$\gamma_1 - 1 + \exp\left(-\rho_c(A)\gamma_1 - \frac{\rho_c(A)n_0}{\gamma_1(n - n_0)}\right) = 0. \tag{6}$$

**Corollary 1.** *Under the same assumptions:*

- *if $\rho_c(A) < 1$,* $$\sigma(A) \leq n_0 + \sqrt{\frac{\rho_c(A)}{1 - \rho_c(A)}} \sqrt{n_0(n - n_0)},$$

- *if $\rho_c(A) \geq 1$,* $$\sigma(A) \leq n - (n - n_0) \exp\left(-\rho_c(A) - \frac{2\rho_c(A)}{\sqrt{4n/n_0 - 3} - 1}\right).$$

*In particular, when $\rho_c(A) < 1$, $\sigma(A) = O(\sqrt{n})$ and the regime is sub-critical.*

The second result (Proposition 2) applies in the case where $A$ is drawn from a uniform distribution over the ensemble of sets of $n_0$ nodes chosen amongst $n$ (denoted as $\mathcal{P}_{n_0}(\mathcal{V})$). This result corresponds to the average-case scenario in a setting where the initial influencer nodes are not known and drawn independently of the transmissions over each edge.

**Proposition 2.** *Define $\rho_c = \rho(\frac{\mathcal{H} + \mathcal{H}^\top}{2})$. Assume the set of influencers $A$ is drawn from a uniform distribution over $\mathcal{P}_{n_0}(\mathcal{V})$. Then, denoting by $\sigma_{uniform}$ the expected number of nodes reached by the cascade starting from $A$:*

$$\sigma_{uniform} \leq n_0 + \gamma_2(n - n_0), \tag{7}$$

*where $\gamma_2$ is the unique solution in $[0, 1]$ of the following equation:*

$$\gamma_2 - 1 + \exp\left(-\rho_c\gamma_2 - \frac{\rho_c n_0}{n - n_0}\right) = 0. \tag{8}$$

**Corollary 2.** *Under the same assumptions:*

- *if $\rho_c < 1$,* $$\sigma_{uniform} \leq \frac{n_0}{1 - \rho_c},$$

- *if $\rho_c \geq 1$,* $$\sigma_{uniform} \leq n - (n - n_0) \exp\left(-\frac{\rho_c}{1 - \frac{n_0}{n}}\right).$$

*In particular, when $\rho_c < 1$, $\sigma_{uniform} = O(1)$ and the regime is sub-critical.*

The difference in the sub-critical regime between $O(\sqrt{n})$ and $O(1)$ for the worst and average case influence is an important feature of our results, and is verified in our experiments (see Sec. 6). Intuitively, when the network is inhomogeneous and contains highly central nodes (e.g. scale-free networks), there will be a significant difference between specifically targeting the most central nodes and random targeting (which will most probably target a peripheral node).

## 4 Application to epidemiology and percolation

Building on the celebrated equivalences between the fields of percolation, epidemiology and influence maximization, we show that our results generalize existing results in these fields.

### 4.1 Susceptible-Infected-Removed (SIR) model in epidemiology

We show here that Proposition 1 further improves results on the SIR model in epidemiology. This widely used model was introduced by Kermac and McKendrick ([14]) in order to model the propagation of a disease in a given population. In this setting, nodes represent individuals, that can be in one of three possible states, susceptible (S), infected (I) or removed (R). At $t = 0$, a subset $A$ of $n_0$ nodes is infected and the epidemic spreads according to the following evolution. Each infected node transmits the infection along its outgoing edge $(i, j) \in \mathcal{E}$ at stochastic rate of occurrence $\beta$ and is removed from the graph at stochastic rate of occurrence $\delta$. The process ends for a given $T > 0$. It is straightforward that, if the removed events are not observed, this infection process is equivalent to $CTIC(\mathcal{F}, T)$ where for any $(i, j) \in \mathcal{E}, f_{ij}(t) = \beta \exp(-\delta t)$. The hazard matrix $\mathcal{H}$ is therefore equal to $\frac{\beta}{\delta}\mathcal{A}$ where $\mathcal{A} = \left(\mathbb{1}_{\{(i,j)\in\mathcal{E}\}}\right)_{ij}$ is the adjacency matrix of the underlying network. Note

that, by Lemma 1, our results can be used in order to model the total number of infected nodes in a setting where infection and recovery rates of a given node exhibit a non-exponential behavior. For instance, incubation periods for different individuals generally follow a log-normal distribution [17], which indicates that continuous-time IC with a log-normal rate of removal might be well-suited to model some kind of infections.

It was recently shown by Draief, Ganesh and Massoulié ([15]) that, in the case of undirected networks, and if $\beta \rho(\mathcal{A}) < \delta$,

$$\sigma(A) \leq \frac{\sqrt{nn_0}}{1 - \frac{\beta}{\delta}\rho(\mathcal{A})}. \tag{9}$$

This result shows, that, when $\rho(\mathcal{H}) = \frac{\beta}{\delta}\rho(\mathcal{A}) < 1$, the influence of set of nodes $A$ is $O(\sqrt{n})$. We show in the next lemma that this result is a direct consequence of Corollary 1: the condition $\rho_c(\mathcal{A}) < 1$ is weaker than $\rho(\mathcal{H}) < 1$ and, under these conditions, the bound of Corollary 1 is tighter.

**Lemma 2.** *For any symmetric adjacency matrix $\mathcal{A}$, initial set of influencers $A$ such that $|A| = n_0 < n$, $\delta > 0$ and $\beta < \frac{\delta}{\rho(\mathcal{A})}$, we have simultaneously $\rho_c(A) \leq \frac{\beta}{\delta}\rho(\mathcal{A})$ and*

$$n_0 + \sqrt{\frac{\rho_c(A)}{1 - \rho_c(A)}} \sqrt{n_0(n - n_0)} \leq \frac{\sqrt{nn_0}}{1 - \frac{\beta}{\delta}\rho(\mathcal{A})}, \tag{10}$$

*where the condition $\beta < \frac{\delta}{\rho(\mathcal{A})}$ imposes that the regime is sub-critical.*

Moreover, these new bounds capture with more accuracy the behavior of the influence in extreme cases. In the limit $\beta \to 0$, the difference between the two bounds is significant, because Proposition 1 yields $\sigma(A) \to n_0$ whereas (9) only ensures $\sigma(A) \leq \sqrt{nn_0}$. When $n = n_0$, Proposition 1 also ensures that $\sigma(A) = n_0$ whereas (9) yields $\sigma(A) \leq \frac{n_0}{1 - \frac{\beta}{\delta}\rho(\mathcal{A})}$. Secondly, Proposition 1 gives also bounds in the case $\beta\rho(\mathcal{A}) \geq \delta$. Finally, Proposition 1 applies to more general cases that the classical homogeneous SIR model, and allows infection and recovery rates to vary across individuals.

## 4.2 Bond percolation

Given a finite undirected graph $\mathcal{G} = (\mathcal{V}, \mathcal{E})$, *bond percolation* theory describes the behavior of connected clusters of the spanning subgraph of $\mathcal{G}$ obtained by retaining a subset $\mathcal{E}' \subset \mathcal{E}$ of edges of $\mathcal{G}$ according to a given distribution. When these removals occur independently along each edge with same probability $1 - p$, this process is called *homogeneous* percolation and is fairly well known (see e.g [18]). The *inhomogeneous* case, where the independent edge removal probabilities $1 - p_{ij}$ vary across the edges, is more intricate and has been the subject of recent studies. In particular, results on critical probabilities and size of the giant component have been obtained by Bollobas, Janson and Riordan in [16]. However, these bounds hold for a particular class of asymptotic graphs (inhomogeneous random graphs) when $n \to \infty$. In the next lemma, we show that our results can be used in order to obtain bounds that hold in expectation for any fixed graph.

**Lemma 3.** *Let $\mathcal{P} = (p_{ij})_{ij} \in [0,1]^{n \times n}$ be a symmetric matrix. Let $\mathcal{G}' = (\mathcal{V}, \mathcal{E}')$ be the undirected subgraph of $\mathcal{G}$ such that each edge $\{i,j\} \in \mathcal{E}$ is removed independently with probability $1 - p_{ij}$. Let $\mathcal{G}_d = (\mathcal{V}, \mathcal{E}_d)$ be the directed graph such that $(i,j) \in \mathcal{E}_d \iff \{i,j\} \in \mathcal{E}$. Then, for any $v \in \mathcal{V}$, the expected size of the connected component containing $v$ in $\mathcal{G}'$ is equal to the influence of $v$ in $\mathcal{G}_d$ under the infection process $DTIC(\mathcal{P})$.*

We now derive an upper bound for $C_1(\mathcal{G}')$, the size of the largest connected component of the spanning subgraph $\mathcal{G}' = (\mathcal{V}, \mathcal{E}')$. In the following, we will denote by $\mathbb{E}[C_1(\mathcal{G}')]$ the expected value of this random variable, given $\mathcal{P} = (p_{ij})_{ij}$.

**Proposition 3.** *Let $\mathcal{G} = (\mathcal{V}, \mathcal{E})$ be an undirected network where each edge $\{i,j\} \in \mathcal{E}$ has an independent probability $1 - p_{ij}$ of being removed. The expected size of the largest connected component of the resulting subgraph $\mathcal{G}'$ is upper bounded by:*

$$\mathbb{E}[C_1(\mathcal{G}')] \leq n\sqrt{\gamma_3}, \tag{11}$$

*where $\gamma_3$ is the unique solution in $[0,1]$ of the following equation:*

$$\gamma_3 - 1 + \frac{n-1}{n} \exp\left(-\frac{n}{n-1}\rho(\mathcal{H})\gamma_3\right) = 0. \tag{12}$$

*Moreover, the resulting network has a probability of being connected upper bounded by:*

$$\mathbb{P}(\mathcal{G}' \text{ is connected}) \leq \gamma_3. \tag{13}$$

In the case $\rho(\mathcal{H}) < 1$, we can further simplify our bounds in the same way than for Propositions 1 and 2.

**Corollary 3.** *In the case $\rho(\mathcal{H}) < 1$, $\mathbb{E}[C_1(\mathcal{G}')] \leq \sqrt{\frac{n}{1-\rho(\mathcal{H})}}$.*

Whereas our results hold for any $n \in \mathbb{N}$, classical results in percolation theory study the asymptotic behavior of sequences of graphs when $n \to \infty$. In order to further compare our results, we therefore consider sequences of spanning subgraphs $(\mathcal{G}'_n)_{n \in \mathbb{N}}$, obtained by removing each edge of graphs of $n$ nodes $(\mathcal{G}_n)_{n \in \mathbb{N}}$ with probability $1 - p_{ij}^n$. A previous result ([16], Corollary 3.2 of section 5) states that, for particular sequences known as *inhomogeneous random graphs* and under a given sub-criticality condition, $C_1(\mathcal{G}'_n) = o(n)$ *asymptotically almost surely* (a.a.s.), i.e with probability going to 1 as $n \to \infty$. Using Proposition 3, we get for our part the following result:

**Corollary 4.** *Assume the sequence $\left(\mathcal{H}^n = \left(-\ln(1-p_{ij}^n)\right)_{ij}\right)_{n \in \mathbb{N}}$ is such that*

$$\limsup_{n\to\infty} \rho(\mathcal{H}^n) < 1. \tag{14}$$

*Then, for any $\epsilon > 0$, we have asymptotically almost surely when $n \to \infty$,*

$$C_1(\mathcal{G}'_n) = o(n^{1/2+\epsilon}). \tag{15}$$

This result is to our knowledge the first to bound the expected size of the largest connected component in general arbitrary networks.

# 5  Application to particular networks

In order to illustrate our theoretical results, we now apply our bounds to three specific networks and compare them to existing results, showing that our bounds are always of the same order than these specific results. We consider three particular networks: 1) star-shaped networks, 2) Erdös-Rényi networks and 3) random graphs with an expected degree distribution. In order to simplify these problems and exploit existing theorems, we will consider in this section that $p_{ij} = p$ is fixed for each edge $\{i,j\} \in \mathcal{E}$. Infection dynamics thus only depend on $p$, the set of influencers $A$, and the structure of the underlying network.

## 5.1  Star-shaped networks

For a star shaped network centered around a given node $v_1$, and $A = \{v_1\}$, the exact influence is computable and writes $\sigma(\{v_1\}) = 1 + p(n-1)$. As $\mathcal{H}(A)_{ij} = -\ln(1-p)\mathbb{1}_{\{i=1,j\neq 1\}}$, the spectral radius is given by

$$\rho\left(\frac{\mathcal{H}(A) + \mathcal{H}(A)^\top}{2}\right) = \frac{-\ln(1-p)}{2}\sqrt{n-1}. \tag{16}$$

Therefore, Proposition 1 states that $\sigma(\{v_1\}) \leq 1 + (n-1)\gamma_1$ where $\gamma_1$ is the solution of equation

$$1 - \gamma_1 = \exp\left(\left(\gamma_1\sqrt{n-1} + \frac{1}{\gamma_1\sqrt{n-1}}\right)\frac{\ln(1-p)}{2}\right). \tag{17}$$

It is worth mentionning that, when $p = \frac{1}{\sqrt{n-1}}$, $\gamma_1 = \frac{1}{\sqrt{n-1}}$ is solution of (17) and therefore the bound is $\sigma(\{v_1\}) \leq 1 + \sqrt{n-1}$ which is tight. Note that, in the case of star-shaped networks, the influence does not present a critical behavior and is always linear with respect to the total number of nodes $n$.

## 5.2  Erdös-Rényi networks

For Erdös-Rényi networks $\mathcal{G}(n,p)$ (*i.e.* an undirected network with $n$ nodes where each couple of nodes $(i,j) \in \mathcal{V}^2$ belongs to $\mathcal{E}$ independently of the others with probability $p$), the exact influence

of a set of nodes is not known. However, percolation theory characterizes the limit behavior of the giant connected component when $n \to \infty$. In the simplest case of Erdös-Rényi networks $\mathcal{G}(n, \frac{c}{n})$ the following result holds:

**Lemma 4.** *(taken from [16]) For a given sequence of Erdös-Rényi networks $\mathcal{G}(n, \frac{c}{n})$, we have:*

- *if $c < 1$, $C_1(\mathcal{G}(n, \frac{c}{n})) \leq \frac{3}{(1-c)^2} \log(n)$ a.a.s.*

- *if $c > 1$, $C_1(\mathcal{G}(n, \frac{c}{n})) = (1 + o(1))\beta n$ a.a.s. where $\beta - 1 + \exp(-\beta c) = 0$.*

As previously stated, our results hold for any given graph, and not only asymptotically. However, we get an asymptotic behavior consistent with the aforementioned result. Indeed, using notations of section 4.2, $\mathcal{H}_{ij}^n = -\ln(1 - \frac{c}{n})\mathbb{1}_{\{i \neq j\}}$ and $\rho(\mathcal{H}^n) = -(n-1)\ln(1 - \frac{c}{n})$. Using Proposition 3, and noting that $\gamma_3 = (1 + o(1))\beta$, we get that, for any $\epsilon > 0$:

- if $c < 1$, $C_1(\mathcal{G}(n, \frac{c}{n})) = o(n^{1/2+\epsilon})$ a.a.s.
- if $c > 1$, $C_1(\mathcal{G}(n, \frac{c}{n})) \leq (1 + o(1))\beta n^{1+\epsilon}$ a.a.s., where $\beta - 1 + \exp(-\beta c) = 0$.

### 5.3 Random graphs with given expected degree distribution

In this section, we apply our bounds to random graphs whose expected degree distribution is fixed (see e.g [19], section 13.2.2). More specifically, let $w = (w_i)_{i \in \{1,...,n\}}$ be the expected degree of each node of the network. For a fixed $w$, let $G(w)$ be a random graph whose edges are selected independently and randomly with probability

$$q_{ij} = \frac{\mathbb{1}_{\{i \neq j\}} w_i w_j}{\sum_k w_k}. \tag{18}$$

For these graphs, results on the *volume* of connected components (i.e the expected sum of degrees of the nodes in these components) were derived in [20] but our work gives to our knowledge the first result on the size of the giant component. Note that Erdös-Rényi $\mathcal{G}(n, p)$ networks are a special case of (18) where $w_i = np$ for any $i \in \mathcal{V}$.

In order to further compare our results, we note that these graphs are also very similar to the widely used *configuration model* where node degrees are fixed to a sequence $w$, the main difference being that the occupation probabilities $p_{ij}$ are in this case not independent anymore. For configuration models, a giant component exists if and only if $\sum_i w_i^2 > 2 \sum_i w_i$ ([21, 22]). In the case of graphs with given expected degree distribution, we retrieve the key role played by the ratio $\sum_i w_i^2 / \sum_i w_i$ in our criterion of non-existence of the giant component given by $\rho(\frac{\mathcal{H}+\mathcal{H}^\top}{2}) < 1$ where

$$\rho\left(\frac{\mathcal{H} + \mathcal{H}^\top}{2}\right) \approx \rho((q_{ij})_{ij}) \leq \frac{\sum_i w_i^2}{\sum_i w_i}. \tag{19}$$

The left-hand approximation is particularly good when the $q_{ij}$ are small. This is for instance the case as soon as there exists $\alpha < 1$ such that, for any $i \in \mathcal{V}$, $w_i = o(n^\alpha)$. The right-hand side is based on the fact that the spectral radius of the matrix $(q_{ij} + \mathbb{1}_{\{i=j\}} w_i^2 / \sum_k w_k)_{ij}$ is given by $\sum_i w_i^2 / \sum_i w_i$.

## 6 Experimental results

In this section, we show that the bounds given in Sec. 3 are tight (i.e. very close to empirical results in particular graphs), and are good approximations of the influence on a large set of random networks. Fig. 1a compares experimental simulations of the influence to the bound derived in proposition 1. The considered networks have $n = 1000$ nodes and are of 6 types (see e.g [19] for further details on these different networks): 1) Erdös-Rényi networks, 2) Preferential attachment networks, 3) Small-world networks, 4) Geometric random networks ([23]), 5) 2D regular grids and 6) totally connected networks with fixed weight $b \in [0, 1]$ except for the ingoing and outgoing edges of the influencer node $A = \{v_1\}$ having weight $a \in [0, 1]$. Except for totally connected networks, edge probabilities are set to the same value $p$ for each edge (this parameter was used to tune the spectral radius $\rho_c(A)$). All points of the plots are averages over 100 simulations. The results show that the bound in proposition 1 is tight (see totally connected networks in Fig. 1a) and close to the real influence for a large

class of random networks. In particular, the tightness of the bound around $\rho_c(A) = 1$ validates the behavior in $\sqrt{n}$ of the worst-case influence in the sub-critical regime. Similarly, Fig. 1b compares

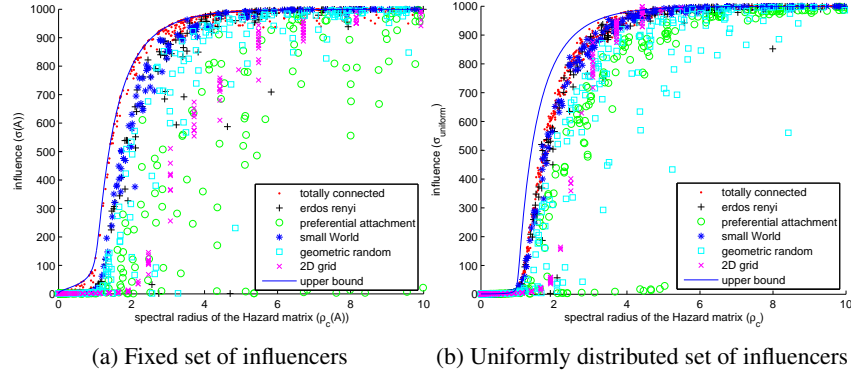

(a) Fixed set of influencers      (b) Uniformly distributed set of influencers

Figure 1: Empirical influence on random networks of various types. The solid lines are the upper bounds in propositions 1 (for Fig. 1a) and 2 (for Fig. 1b).

experimental simulations of the influence to the bound derived in proposition 2 in the case of random initial influencers. While this bound is not as tight as the previous one, the behavior of the bound agrees with experimental simulations, and proves a relatively good approximation of the influence under a random set of initial influencers. It is worth mentioning that the bound is tight for the sub-critical regime and shows that corollary 2 is a good approximation of $\sigma_{\text{uniform}}$ when $\rho_c < 1$. In order to verify the criticality of $\rho_c(A) = 1$, we compared the behavior of $\sigma(A)$ w.r.t the size of the network $n$. When $\rho_c(A) < 1$ (see Fig. 2a in which $\rho_c(A) = 0.5$), $\sigma(A) = O(\sqrt{n})$, and the bound is tight. On the contrary, when $\rho_c(A) > 1$ (see Fig. 2b in which $\rho_c(A) = 1.5$), $\sigma(A) = O(n)$, and $\sigma(A)$ is linear w.r.t. $n$ for most random networks.

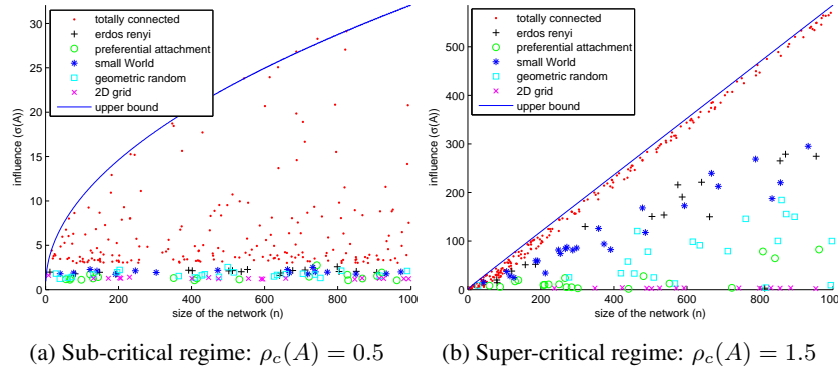

(a) Sub-critical regime: $\rho_c(A) = 0.5$      (b) Super-critical regime: $\rho_c(A) = 1.5$

Figure 2: Influence w.r.t. the size of the network in the sub-critical and super-critical regime. The solid line is the upper bound in proposition 1. Note the square-root versus linear behavior.

# 7 Conclusion

In this paper, we derived the first upper bounds for the influence of a given set of nodes in any finite graph under the Independent Cascade Model (ICM) framework, and relate them to the spectral radius of a given *hazard matrix*. We show that these bounds can also be used to generalize previous results in the fields of epidemiology and percolation. Finally, we provide empirical evidence that these bounds are close to the best possible for general graphs.

### Acknowledgments

This research is part of the SODATECH project funded by the French Government within the program of "*Investments for the Future – Big Data*".

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
