[Supplementary Material]

## Mathematical Arguments

**Proof of Lemma 1**

We prove here the equivalence of propagation dynamics $DTIC(\mathcal{P}), CTIC(\mathcal{F}, \infty)$ and $RN(\mathcal{P})$, provided that for any $(i,j) \in \mathcal{E}$, $\int_0^\infty f_{ij}(t)dt = \mathcal{H}_{ij}$. More specifically, we prove the following lemma, that will be useful in the subsequent proofs. In the following, we will denote by $X_i$ the state of node $i$ at the end of the infection process, i.e $X_i = 1$ if the infection has reached node $i$, and $X_i = 0$ otherwise.

**Lemma 5.** *Let $\mathcal{G} = (\mathcal{V}, \mathcal{E})$ be a given directed network and $A \subset \mathcal{V}$ a set of influencers. For any $i \notin A$, we denote by $\mathcal{Q}_i$ the collection of directed paths (without loops) in $\mathcal{G}$ from $A$ to node $i$. Then, under the infection processes $DTIC(\mathcal{P}), CTIC(\mathcal{F}, \infty)$ and $RN(\mathcal{P})$, we have $\forall i \notin A$,*

$$X_i = 1 - \prod_{q \in \mathcal{Q}_i} \left(1 - \prod_{(j,l) \in q} E_{jl}\right), \tag{1}$$

*where the $(E_{jl})_{jl}$ are independant Bernoulli random variables $E_{jl} \sim \mathcal{B}(p_{jl})$ for infection processes $DTIC(\mathcal{P})$ and $RN(\mathcal{P})$, and $E_{jl} \sim \mathcal{B}\left(1 - \exp(-\int_0^\infty f_{jl}(t)dt)\right)$ for infection process $CTIC(\mathcal{F}, \infty)$.*

*Proof.* First, note that, for $RN(\mathcal{P})$, the random variables $1_{\{(j,l)\in\mathcal{E}'\}}$ and,for $DTIC(\mathcal{P})$, the indicator function of the events that node $j$ succeeds in infecting node $l$ if $j$ is infected during the process and $l$ is still healthy at that time are independant Bernoulli variables $E_{jl} \sim \mathcal{B}(p_{jl})$ and can all be drawn at $t = 0$. Moreover, by definition of the infection processes, a node $i \in \mathcal{V}$ is reached by the contagion if and only if there exists a path from $A$ to $i$, such that each of its edges transmitted the contagion. We thus have for $DTIC(\mathcal{P})$ and $RN(\mathcal{P})$:

$$X_i = 1 - \prod_{q \in \mathcal{Q}_i} \left(1 - \prod_{(j,l) \in q} E_{jl}\right). \tag{2}$$

For $CTIC(\mathcal{F}, \infty)$, the variables drawn at the beginning of the infection process are the (possibly infinite) times $\tau_{jl}$ such that node $j$ will infect node $l$ at time $t_j + \tau_{jl}$ if node $j$ has been infected at time $t_j$, and node $l$ has not been infected by another node before time $t_j + \tau_{jl}$. By definition, these independent random variables have the following survival function:

$$P(\tau_{jl} < t) = 1 - \exp\left(-\int_0^t f_{jl}(s)ds\right) \tag{3}$$

Therefore, we have by the same arguments than previously,

$$X_i = 1 - \prod_{q \in \mathcal{Q}_i} \left(1 - \prod_{(j,l) \in q} 1_{\{\tau_{jl} < \infty\}}\right), \tag{4}$$

which proves the result for $CTIC(\mathcal{F}, \infty)$, defining $E_{jl} = 1_{\{\tau_{jl} < \infty\}}$ □

Lemma 1 is then a direct corollary of Lemma 5 in the case where, for any $(j,l) \in \mathcal{E}$, $\int_0^\infty f_{jl}(t)dt = \mathcal{H}_{jl}$.

**Proofs of Proposition 1 and Corollary 1**

We develop here the full proofs for Proposition 1 and Corollary 1 that apply to any set of initially infected nodes. We will first need to prove two useful results: Lemma 6, that proves for $j \in \mathcal{V}$ a positive correlation between the events 'node $j$ did not infect node $i$ during the epidemic' and Lemma 8, that bound the probability that a given node gets infected during the infection process.

**Lemma 6.** $\forall i \notin A$, $\{1 - X_j E_{ji}\}_{j \in \mathcal{V}}$ *are positively correlated.*

*Proof.* We will make use of the FKG inequality ([1]):

**Lemma 7.** *(FKG inequality) Let $L$ be a finite distributive lattice, and $\mu$ a nonnegative function on $L$, such that, for any $(x, y) \in L^2$,*

$$\mu(x \vee y)\mu(x \wedge y) \leq \mu(x)\mu(y) \tag{5}$$

*Then, for any non-decreasing function $f$ and $g$ on $L$*

$$\left(\sum_{x \in L} f(x)g(x)\right)\left(\sum_{x \in L} \mu(x)\right) \geq \left(\sum_{x \in L} f(x)\mu(x)\right)\left(\sum_{x \in L} g(x)\mu(x)\right) \tag{6}$$

For a given set of influencers $A$, the $X_j$ are deterministic functions of the independent random variables $(E_{ij})_{ij}$. Thus, let $f_{ij}(\{E_{i'j'}\}_{(i',j')}) = 1 - X_j E_{ji}$. In order to apply the FKG inequality, we first need to show that each $f_{ij} : \{0, 1\}^{n^2} \to \{0, 1\}$ is decreasing with respect to the natural partial order on $\{0, 1\}^{n^2}$ (i.e. $X \leq Y$ if $X_i \leq Y_i$ for all $i$). Let $u \in \{0, 1\}^{n^2}$ be a given transmission state of the edges of the network. In order to prove the decreasing behavior of $f_{ij}$, it is sufficient to show that $f_{ij}(u)$ is decreasing with respect to every $u_{(i,j)}$.

But from Lemma 5, it is obvious that $X_i(u) = 1 - \prod_{q \in \mathcal{Q}_i}(1 - \prod_{(j,l) \in q} u_{(j,l)})$ is increasing with respect to every $u_{(i,j)}$. This implies that $f_{ij}(u) = 1 - X_j(u)u_{(j,i)}$ is decreasing with respect to every $u_{(i,j)}$ and that $f_{ij} : \{0, 1\}^{n^2} \to \{0, 1\}$ is decreasing with respect to the natural partial order on $\{0, 1\}^{n^2}$.

Finally, since we consider a product measure (due to the independence of the $E_{ij}$) on a product space, we can apply the FKG inequality to $\{1 - X_j E_{ji}\}_{j \in \{1, \dots, N\}}$, and these random variables are positively correlated. $\qquad\square$

The next lemma ensures that the variables $X_i$ satisfy an implicit inequation that will be the starting point of the proof of Proposition 1.

**Lemma 8.** *For any $A$ such that $|A| = n_0 < n$ and for any $i \notin A$, the probability $\mathbb{E}[X_i]$ that node $i$ will be reached by the contagion originating from $A$ verifies:*

$$\mathbb{E}[X_i] \leq 1 - \exp\left(-\sum_j \mathcal{H}_{ji}\mathbb{E}[X_j]\right) \tag{7}$$

*Proof.* We first note that a node is infected if and only if one of its neighbors is infected, and the respective ingoing edge transmitted the contagion. Thus

$$X_i = 0 \Leftrightarrow \forall j \in \{1, \dots, n\}, X_j = 0 \text{ or } E_{ji} = 0, \tag{8}$$

which implies the following alternative expression for $X_i$:

$$1 - X_i = \prod_j (1 - X_j E_{ji}). \tag{9}$$

Moreover, the positive correlation of $\{1 - X_j E_{ji}\}_{j \in \{1, \dots, N\}}$ implies that

$$\mathbb{E}[\prod_j (1 - X_j E_{ji})] \geq \prod_j \mathbb{E}[1 - X_j E_{ji}] \tag{10}$$

which leads to

$$
\begin{aligned}
\mathbb{E}[X_i] \ &\leq 1 - \prod_j \mathbb{E}[1 - X_j E_{ji}] \\
&= 1 - \prod_j (1 - \mathbb{E}[X_j]\mathbb{E}[E_{ji}]) \\
&= 1 - \exp\left(\sum_j \ln(1 - \mathbb{E}[X_j]\mathbb{E}[E_{ji}])\right) \\
&\leq 1 - \exp\left(\sum_j \ln(1 - \mathbb{E}[E_{ji}])\mathbb{E}[X_j]\right) \\
&= 1 - \exp\left(-\sum_j \mathcal{H}_{ji}\mathbb{E}[X_j]\right)
\end{aligned}
\tag{11}
$$

since we have on the one hand, for any $x \in [0, 1]$ and $a < 1$, $\ln(1 - ax) \geq \ln(1 - a)x$, and on the other hand $\mathbb{E}[E_{ji}] = 1 - \exp(-\mathcal{H}_{ji})$ by definition of $\mathcal{H}$. $\qquad\square$

Using Lemma 8, we are now ready to start the proof of Proposition 1.

*Proof of Proposition 1.* In order to simplify notations, we define $Z_i = (\mathbb{E}[X_i])_i$ that we collect in the vector $Z = (Z_i)_{i \in [1...n]}$. Using lemma 8 and convexity of exponential function, we have for any $u \in R^n$ such that $\forall i \in A, u_i = 0$ and $\forall i \notin A, u_i \geq 0$,

$$u^\top Z \leq |u|_1 \left( 1 - \sum_{i=1}^{n-1} \frac{u_i}{|u|_1} \exp(-(\mathcal{H}^\top Z)_i) \right) \leq |u|_1 \left( 1 - \exp\left( - \frac{Z^\top \mathcal{H} u}{|u|_1} \right) \right) \tag{12}$$

where $|u|_1 = \sum_i |u_i|$ is the $L_1$-norm of $u$.

Now taking $u = (1_{i \notin A} Z_i)_i$ and noting that $\forall i \in \{1, \ldots, n\}, \forall j \in A, \mathcal{H}(A)_{ij} = 0$, we have

$$\frac{Z^\top Z - n_0}{|Z|_1 - n_0} \leq 1 - \exp\left( - \frac{Z^\top \mathcal{H}(A) Z}{|Z|_1 - n_0} \right) \leq 1 - \exp\left( - \frac{\rho_c(A)(Z^\top Z - n_0)}{|Z|_1 - n_0} - \frac{\rho_c(A) n_0}{|Z|_1 - n_0} \right) \tag{13}$$

where $\rho_c(A) = \rho\left( \frac{\mathcal{H}(A) + \mathcal{H}(A)^\top}{2} \right)$. Defining $y = \frac{Z^\top Z - n_0}{|Z|_1 - n_0}$ and $z = |Z|_1 - n_0 = \sigma(A) - n_0$, the aforementioned inequation rewrites

$$y \leq 1 - \exp\left( - \rho_c(A) y - \frac{\rho_c(A) n_0}{z} \right) \tag{14}$$

But by Cauchy-Schwarz inequality applied to $u$, $(n - n_0)(Z^\top Z - n_0) \geq (|Z|_1 - n_0)^2$, which means that $z \leq y(n - n_0)$. We now consider the equation

$$x - 1 + \exp\left( - \rho_c(A) x - \frac{\rho_c(A) n_0}{x(n - n_0)} \right) = 0 \tag{15}$$

Because the function $f : x \to x - 1 + \exp\left( - \rho_c(A) x + \frac{\rho_c(A) n_0}{x(n - n_0)} \right)$ is continuous, verifies $f(1) > 0$ and $\lim_{x \to 0^+} f(x) = -1$, equation 15 admits a solution $\gamma_1$ in $]0, 1[$.

We then prove by contradiction that $z \leq \gamma_1(n - n_0)$. Let us assume $z > \gamma_1(n - n_0)$. Then $y \leq 1 - \exp\left( - \rho_c(A) y - \frac{\rho_c(A) n_0}{\gamma_1(n - n_0)} \right)$. But the function $h : x \to x - 1 + \exp\left( - \rho_c(A) x + \frac{\rho_c(A) n_0}{\gamma_1(n - n_0)} \right)$ is convex and verifies $h(0) < 0$ and $h(\gamma_1) = 0$. Therefore, for any $y > \gamma_1$, $0 = f(\gamma_1) \leq \frac{\gamma_1}{y} f(y) + (1 - \frac{\gamma_1}{y}) f(0)$, and therefore $f(y) > 0$. Thus, $y \leq \gamma_1$. But $z \leq y(n - n_0) \leq \gamma_1(n - n_0)$ which yields the contradiction. $\qquad \square$

*Proof of Corollary 1.* We distinguish between the cases $\rho_c(A) > 1$ and $\rho_c(A) \leq 1$.

**Case $\rho_c(A) < 1$.** Using Eq. 15 and the fact that $\exp(z) \geq 1 + z$, we get $\gamma_1 \leq \rho_c(A) \gamma_1 + \frac{\rho_c(A) n_0}{\gamma_1(n - n_0)}$ which rewrites $\gamma_1 \leq \sqrt{\frac{\rho_c(A) n_0}{(1 - \rho_c(A))(n - n_0)}}$ in the case $\rho_c < 1$. Therefore,

$$\sigma(A) \leq n_0 + \sqrt{\frac{\rho_c(A)}{1 - \rho_c(A)}} \sqrt{n_0(n - n_0)} \tag{16}$$

**Case $\rho_c(A) \geq 1$.** Using Eq. 15, we get $\gamma_1 - 1 + \exp(-\frac{\rho_c(A) n_0}{\gamma_1(n - n_0)}) \geq 0$, which implies $\gamma_1 \ln(\frac{1}{1 - \gamma_1}) \geq \frac{\rho_c(A) n_0}{n - n_0} \geq \frac{n_0}{n - n_0}$. By concavity of the logarithm, we therefore have $\gamma_1^2 \geq \frac{n_0(1 - \gamma_1)}{n - n_0}$ which means that $\gamma_1(n - n_0) \geq \frac{n_0(\sqrt{4n/n_0 - 3} - 1)}{2}$. By plugging this lower bound in Eq. 15, we obtain

$$\sigma(A) \leq n_0 + \left( 1 - \exp\left( - \rho_c(A) - \frac{2\rho_c(A)}{\sqrt{4n/n_0 - 3} - 1} \right) \right)(n - n_0) \tag{17}$$

$\qquad \square$

**Proofs of Proposition 2 and Corollary 2**

In this subsection, we develop the proofs for Proposition 2 and Corollary 2 in the case when the set of initially infected node is drawn from a uniform distribution over $\mathcal{P}_{n_0}(\mathcal{V})$.

We start with an important lemma that will play the same role in the proof of Proposition 2 than Lemma 8 in the proof of Proposition 1.

**Lemma 9.** *Define $\rho_c = \rho(\frac{\mathcal{H}+\mathcal{H}^\top}{2})$. Assume $A$ is drawn from an uniform distribution over $\mathcal{P}_{n_0}(\mathcal{V})$. Then, for any $i \in \mathcal{V}$, the probability $\mathbb{E}[X_i]$ that node $i$ will be reached by the contagion satisfies the following implicit inequation:*

$$\mathbb{E}[X_i] \leq 1 - \frac{n - n_0}{n} \exp\left(-\frac{n}{n - n_0}\sum_j \mathcal{H}_{ji}\mathbb{E}[X_i]\right) \tag{18}$$

*Proof.*

$$
\begin{aligned}
\mathbb{E}[X_i] &= \mathbb{E}[1_{\{i \in A\}}] + \mathbb{E}[1_{\{i \notin A\}}]\mathbb{E}[\mathbb{E}[X_i|A]|i \notin A] \\
&\leq \tfrac{n_0}{n} + \tfrac{n-n_0}{n}\left(1 - \mathbb{E}[\exp\left(-\sum_j \mathcal{H}_{ji}\mathbb{E}[X_j|A]\right)|i \notin A]\right) \\
&\leq \tfrac{n_0}{n} + \tfrac{n-n_0}{n}\left(1 - \exp\left(-\mathbb{E}[\sum_j \mathcal{H}_{ji}\mathbb{E}[X_j|A]|i \notin A]\right)\right) \\
&= 1 - \tfrac{n-n_0}{n}\exp\left(-\sum_j \mathcal{H}_{ji}\mathbb{E}[X_j|i \notin A]\right) \\
&\leq 1 - \tfrac{n-n_0}{n}\exp\left(-\tfrac{n}{n-n_0}\sum_j \mathcal{H}_{ji}\mathbb{E}[X_j]\right)
\end{aligned}
\tag{19}
$$

where the first inequality is Lemma 8 and the second one is Jensen inequality for conditional expectations. $\square$

*Proof of Proposition 2.* We define $Z_i = (\mathbb{E}[X_i])_i$ that we collect in the vector $Z = (Z_i)_{i \in [1...n]}$. Then, using Lemma 9, and convexity of exponential function, we have:

$$\frac{Z^\top Z}{|Z|_1} \leq \left(1 - \frac{n-n_0}{n}\sum_{i=1}^n \frac{Z_i}{|Z|_1}\exp\left(-\frac{n}{n-n_0}(\mathcal{H}^\top Z)_i\right)\right) \leq \left(1 - \frac{n-n_0}{n}\exp\left(-\frac{n}{n-n_0}\frac{Z^\top \mathcal{H} Z}{|Z|_1}\right)\right) \tag{20}$$

Now, defining $y = \frac{Z^\top Z}{|Z|_1}$, we have by Cauchy-Schwarz inequality $|Z|_1 \leq ny$ where $y \leq 1 - \frac{n-n_0}{n}\exp\left(-\frac{n}{n-n_0}\rho_c y\right)$. Because function $f : x \to x - 1 + \frac{n-n_0}{n}\exp\left(-\frac{n}{n-n_0}\rho_c y\right)$ is continuous and convex over $]0,1[$, $f(0) < 0$ and $f(1) > 0$, there exists a solution $\gamma \in ]0,1[$ of the equation $f(x) = 0$. By the same arguments than in proof of Proposition 1, we have that, for any $z \in [0,1]$, $f(z) \leq 0 \Rightarrow z \leq \gamma$. This proves the uniqueness of $\gamma$ as well as the fact that $y \leq \gamma$. Now, defining $\gamma_2 = \frac{n_0}{n} + \frac{n-n_0}{n}\gamma$, we have on the one hand

$$\sigma_{\text{uniform}} \leq n_0 + \gamma_2(n - n_0) \tag{21}$$

and on the other hand

$$\gamma_2 - 1 + \exp\left(-\rho_c\gamma_2 - \frac{\rho_c n_0}{n - n_0}\right) = 0 \tag{22}$$

which proves the proposition. $\square$

*Proof of Corollary 2.* In the case $\rho_c < 1$, using Proposition 2 and the fact that $\exp(z) \geq 1 + z$, we get $\gamma_2 \leq \rho_c\gamma_2 + \frac{\rho_c n_0}{n-n_0}$ which rewrites $\gamma_2 \leq \frac{\rho_c n_0}{(1-\rho_c)(n-n_0)}$ in the case $\rho_c < 1$. Therefore,

$$\sigma_{\text{uniform}} \leq n_0\left(1 + \frac{\rho_c}{1 - \rho_c}\right) = \frac{n_0}{1 - \rho_c} \tag{23}$$

The second claim is straightforward from Proposition 2, using the fact that $\gamma_2 \leq 1$. $\square$

**Proofs of Lemma 2, Lemma 3, Proposition 3 and Corollary 4**

*Proof of Lemma 2.* Because matrices $\frac{\mathcal{H}(A)+\mathcal{H}(A)^\top}{2}$ and $\frac{\beta}{\delta}\mathcal{A}$ are symmetric and verify $0 \leq \frac{\mathcal{H}(A)+\mathcal{H}(A)^\top}{2} \leq \frac{\beta}{\delta}\mathcal{A} = \mathcal{H}$ where $\leq$ stands for the coefficient-wise inequality, we have $\rho(\frac{\mathcal{H}(A)+\mathcal{H}(A)^\top}{2}) \leq \frac{\beta}{\delta}\rho(\mathcal{A})$ as a direct consequence of the Perron-Frobenius theorem (see e.g [2]). We now introduce the function

$$f : \rho \to n_0 + \sqrt{\frac{\rho}{1-\rho}}\sqrt{n_0(n-n_0)} - \frac{\sqrt{nn_0}}{1-\rho}$$

We have $f(0) < 0$ and $f'(\rho) = \sqrt{n_0(n-n_0)}\frac{\rho}{(1-\rho)^{3/2}} - \sqrt{n_0 n}\frac{1}{(1-\rho)^2} < 0$. Therefore, $f(\rho) < 0$ for any $\rho \in [0,1]$, which proves the Lemma. $\qquad\square$

*Proof of Lemma 3.* First, note that, for bond percolation, the random variables $1_{\{\{j,l\}\in\mathcal{E}'\}}$ are independent Bernoulli variables $F_{\{j,l\}} \sim \mathcal{B}(p_{jl})$. We therefore have, similarly than in the proof of Lemma 5

$$X_i = 1 - \prod_{q\in\mathcal{Q}_i}\left(1 - \prod_{\{j,l\}\in q}F_{\{j,l\}}\right). \tag{24}$$

where $X_i$ is 1 if node $i$ belongs to the connected component containing the influencer node $v$, and is 0 otherwise. We then show that, because $\mathcal{P}$ is symmetric, for any infection process $DTIC(\mathcal{P})$ on the directed graph $\mathcal{G}_d$, we can also define independent variables $F'_{\{j,l\}} \sim \mathcal{B}(p_{jl})$ such that the final infection state $X'_i$ of node $i$ is:

$$X'_i = 1 - \prod_{q\in\mathcal{Q}_i}\left(1 - \prod_{\{j,l\}\in q}F'_{\{j,l\}}\right), \tag{25}$$

which proves that $X_i$ and $X'_i$ have the same probability distribution.

Indeed, the event that node $j$ makes an attempt to infect node $l$ will never occur in the same epidemic than the event that node $l$ makes an attempt to infect node $j$. Therefore, drawing two variables $E_{jl}$ and $E_{lj}$ at the beginning of each epidemic and letting the dynamic decide which of the two results will be used, or drawing only one variable $F'_{\{j,l\}} \sim \mathcal{B}(p_{jl})$ and using it for each epidemic to decide wether the infection can spread along the edge $\{j,l\}$ or not is strictly equivalent, given that $E_{jl}$ and $E_{lj}$ are independent and have the same distribution. From equations 24 and 25, we see that, for any $i \in \mathcal{V}$, the probability that a node $i$ is infected is the same for the two processes. $\qquad\square$

*Proof of Proposition 3.* By proposition 2 applied to the case $n_0 = 1$ with the notation $\gamma_3 = \frac{(n-1)\gamma_2+1}{n}$, we get $\sigma_{\text{uniform}} \leq n\gamma_3$. We then use the fact that, when the influencer node is uniformly randomly drawn on $\mathcal{V}$, it belongs to the largest connected component and therefore creates an infection of $C_1(\mathcal{G}')$ nodes with probability $\frac{C_1(\mathcal{G}')}{n}$. Therefore, $\mathbb{E}[\frac{C_1(\mathcal{G}')}{n}C_1(\mathcal{G}')] \leq \sigma_{\text{uniform}} \leq n\gamma_3$. But $\mathbb{E}[C_1(\mathcal{G}')^2] \geq \mathbb{E}[C_1(\mathcal{G}')]^2$ which yields $\mathbb{E}[C_1(\mathcal{G}')] \leq n\sqrt{\gamma_3}$. Moreover, denoting as $C_A(\mathcal{G}')$ the size of the connected component containing the influencer node, we have $\sigma_{\text{uniform}} = \mathbb{E}[C_A(\mathcal{G}')] = \sum_i i\mathbb{P}(C_A(\mathcal{G}') = i) \geq n\mathbb{P}(C_A(\mathcal{G}') = n) = n\mathbb{P}(\mathcal{G}' \text{ is connected})$, and therefore $\mathbb{P}(\mathcal{G}' \text{ is connected}) \leq \gamma_3$. $\qquad\square$

*Proof of Corollary 4.* According to Eq. 15 of the article, there exists $m \in \mathbb{N}$ and $\eta < 1$ such that for any $n \geq m$, $\rho(\mathcal{H}^n) \leq \eta$. Therefore, Corrolary 3 implies $\mathbb{E}[C_1(\mathcal{G}'_n)] \leq \sqrt{\frac{n}{1-\eta}}$. But for any $\delta > 0$, $\mathbb{P}(C_1(\mathcal{G}'_n) > \delta n^{1/2+\epsilon}) \leq \frac{\mathbb{E}[C_1(\mathcal{G}'_n)]}{\delta n^{1/2+\epsilon}} = o(1)$ which proves the corollary. $\qquad\square$

## Additional references

[1] Cees M Fortuin, Pieter W Kasteleyn, and Jean Ginibre. Correlation inequalities on some partially ordered sets. *Communications in Mathematical Physics*, 22(2):89–103, 1971.

[2] Carl D Meyer. *Matrix analysis and applied linear algebra*, volume 2. Siam, 2000.