[Reviews · NeurIPS 2014]

Submitted by Assigned_Reviewer_6

This paper presents a bound on the "influence" from a given (or random)
set of nodes on a graph, under a number of popular influence propagation
models. The bound is in terms of the largest eigenvalue of the hazard
matrix. I read the accompanying proofs and they seem reasonable, although
I did not check them completely. The paper compares the bounds to existing
bounds in epidemiology and percolation and shows ways in which the new
bounds are more general or better. Finally, experimental results are given
showing the tightness (or slack) in the bounds for randomly generated networks
from a variety of random network models.

The paper appears to have been written in haste and the notation is not as
clean as desired. Some quantities (script-A as an adjacency matrix) are
reintroduced. Others (see below) are not given the best notation. The proofs
are only given in supplementary material (yet, they are critical to be
reviewed and understood and the bulk of the novelty).

The work, if one is interested in such results, is useful, good, and,
I believe, correct. While the presentation could be improved, the
intellectual contribution is good and worthy of publication.

My main concern (aside from the presentation) is the suitability for
the NIPS audience. NIPS has always cast a wide net and this paper may
well fall within the net, especially given current trends. However, it
relationship to "learning" is nebulous, and the results demonstrate the
bound is not close to being tight for preferential attachment networks.
I leave this decision to the program committee, however. Personally,
I found the paper interesting.

specific comments:
pf of Proposition 1: I don't follow equation 13. In particular, I'm uncertain
what the notation \rho_c(Z^TZ - n_0) means. \rho_c appears to only apply
to A (or at least only to a matrix, if we drop the _c part). Z^TZ - n_0 is
a scalar, I think. In general, the steps around this area appear to have been
written in a haste and perhaps not presented as well as possible.

Why is \rho_c not something more related to its form, like \rho_H to reminder
the reader that the argument is "pushed through" H first?

Corollary 2 uses \rho_c instead of \rho_c(A); I don't think there is a
distinction.

Just before Lemma 2: Having H not be the same as H(A) is very confusing.
It isn't clear at first why \rho_c(A) is not the same as \rho(H).

In the experimental results, the "preferential attachment" networks are
quite far from the bound, and rather random, suggesting that \rho_c(A) is
not a good characterization of them. This is a very popular social network
model (social networks being the motivating factor of the introduction).
Summary: The paper, although it could be presented better, presents new results
bounding the influence of propagation in a network. The paper is worthy of
publication; my only concern is whether NIPS is the right venue (although
I think it probably is).

Submitted by Assigned_Reviewer_33

The authors present novel theoretical upper bounds on the total number
of nodes that are eventually activated/influenced in the Indepnedent
Cascade (IC) model in terms of spectral properties of the "hazard
matrix" (defined in terms of the edge probabilities). They first
present a worst case bound where the initial influencers are chosen to
maximize the influence, and then present an average case bound where
the initially influencers are chosen at random. In each case,
the bounds demonstrate sub-critical and super-critical behavior. In
particular, the sub-critical regime occurs when the spectral radius of
the appropriate matrix is less than one, and gives bounds of
O(sqrt(n)) infected nodes in the worst case and O(1) in the average
case. The show how their general results can apply to improve
results on SIR epidemic models and characterizing giant components in
random graphs. They also give results for particular families of
graphs and demonstrate the accuracy of their bounds empirically.

The analysis of the paper is sophisticated and rigorous and gives
detailed upper bounds that seem to match or beat existing bounds in a
number of areas. The paper is well-written and clear.

While the flavor of the results is not incredibly surprising---there
are many similar results relating giant components, epidemic spread,
etc. to the spectral radius of some matrix---these results are
significant because of their tightness, broad applicability, and
quality of the analysis. They could potentially have algorithmic
significance by providing upper bounds for the influence maximization
problem (provided that the upper bounds are tighter than those given
by the approximation guarantee of the algorithm).

The paper is atypical for NIPS in that it is a theoretical analysis of
a mathematical model without a real algorithmic component. Since it
sheds light on models that are actively being researched by the NIPS
community and makes interesting connections between those and other
models I believe it is within scope.

Comments / questions:

* I believe the statement of Lemma 1 can be strenghtened to say that
the distribution of the final set of infected nodes is identical
under each of these processes (not just that the marginal
probability of each node i being infected is the same).

* I think there is a minor problem with the proof of Lemma 1 in the
supplementary material. Eq. (1) gives a system of equations that
must be satisfied by the random variables {X_i}, but it does not
uniquely define the random process because there are also other
solutions to the set of equations. In particular, one can set
X_i = 1 for all nodes in some cycle that is *not* reachable from
from A and still satisfy these equations. So, showing that all three
processes give random variables that satisfy these equations doesn't
prove their equivalence. It should be easy to fix the proof, e.g.,
by using Eq. (8) instead as the characterization of the process.

* The difference between O(sqrt(n)) and O(1) in the sub-critical
regimes for worst and average cases is interesting. Can you provide
any more intuition for this? Are there any other phase transitions
in the average case behavior?

* Can you say anything about lower bounds in the super-critical cases?

* The applicability of DTIC to undirected graphs in Lemma 3 was a
minor confusion DTIC was defined for directed graphs. The
supplementary material makes it clear that this this can be
interpreted as DTIC in the bidirected graph where p_{ij} =
p_{ji}. Perhaps this can be clarified in the main paper.

* In the experiments, how are the edge probabilities set? What
parameter(s) are tuned to vary the spectral radius? More details
would help understand the experiments better.

* Is each point in your plots the result of a single simulation or the
average over multiple trials? An average over multiple trials would
be a cleaner match to your bounds on the expected value. Error bars
would also help illustrate the variability. On that note (assuming
data points are individual trials), in Figure 3(a) it looks like the
variance of the number of infected nodes is higher in the
sub-critical regime. Can you say anything about this theoretically?

Summary: This is a strong theoretical paper that gives spectral bounds on the
expected number of nodes influenced in the Independent Cascade model
and demonstrates the applicability of these bounds the SIR model in
epidemiology and giant components in random graphs.

Submitted by Assigned_Reviewer_35

Paper Summary:
This paper provides new theoretical bounds for the long-term influence of a node under the well-known independent-cascade model (ICM), with applications to different types of networks and with experimental results on simulated networks that confirm the theoretical results.

Quality
Note to authors: this paper is somewhat outside my area of expertise, so I apologize for that in advance and as a consequence will keep my comments brief and somewhat high level. In addition I have gives a low weight under "confidence" on the review form so that the reviews of the other reviewers (who are hopefully more informed about the topic) can be given more weight.

With the caveat above, I liked the paper. The problem is of broad interest and the authors appear to do have some interesting new results here. The bounds that they derive appear to be applicable to a fairly wide range of networks (see Section 5) and to provide some nice insights about influence propagation in these different networks. Section 6 on simulation results provides confirmation of the theoretical results across a range of simulated network types: although this section is fairly short this seems fine for a paper that is primarily theoretical in nature.

Originality
As far as I can tell the work appears to be original. I like the idea of the hazard matrix, which appears (according to the authors at least) to be novel.

Significance
As far as I can tell the paper would appear to have potentially broad impact.

Clarity
The paper is well-written and clear - it was easily the best-written paper in my batch of 4 papers to review (although the bar was set pretty low by the other papers).
Summary: The paper appears to be a nice contribution to the set of known theoretical results for influence maximization in network models and would appear to worth considering for acceptance (note however that this is an educated guess on my part).
Author Feedback
Author rebuttal: We thank the reviewers for their positive evaluation and their useful comments. Our reply to the main points they have raised follows:

1. Improvement of Lemma 1 and its proof:

We agree with Reviewer 33 that the result should be strengthened in the paper “to say that the distribution of the final set of infected nodes is identical under each of these processes”, although we only used the fact that the marginal probabilities are equal in the rest of the paper. We also agree that the proof of Lemma 5 should rely on equation (8) and not equation (1). The arguments in the proof of Lemma 5 actually prove equation (8) for all processes which uniquely defines the infection behavior. We will make this minor correction in the final version of the paper if accepted.

2. Insights on the different behavior between worst-case (O(sqrt(n))) and average-case behavior (O(1)):

A good example of this behavior is a star-shaped network of size n with transition probabilities 1/sqrt(n-1). When influence spreads from the central node, the bound in Prop. 1 is tight and gives the true value of influence (see section 5.1): 1+sqrt(n-1). However, when influence spreads from a peripheral node, influence is 1+(1/sqrt(n-1))*(1+(n-2)/sqrt(n-1))=2+o(1). Therefore, the overall average-case influence is O(sqrt(n)/n+1)=O(1). This illustrates the following intuition: when the network is inhomogeneous and contains highly central nodes (e.g. scale-free networks), there will be a significant difference between specifically targeting the most central nodes and random targeting (which will most probably target a peripheral node). A possible experiment in order to validate this behavior would be to compute the worst case influence over preferential attachment networks, instead of targeting a node at random as in our experiments (figures 1 and 3).

3. On lower bounds in the super-critical case:

We have reached the following conclusions: a) upper bounds on influence can rely on global features of the graph: wherever the infection starts, it will not spread to many nodes if the graph is not sufficiently infectious and connected “on average”, b) however, lower bounds depend on the local structure of the zone where the infection starts. For instance, a complete sub-graph of size sqrt(n) with n-sqrt(n) isolated nodes will have O(1) average influence for any value of rho_c. Therefore, lower bounds usually depend on isoperimetric constants eta(m), that basically control the minimum number of outgoing edges for any subset of at most m nodes in the graph (see e.g (Draief et al., 2006), reference [15] of our paper). But as the cornerstones of our results are Lemmas 8 and 9 which do not easily express with respect to these constants we think that our approach is not suited for the derivation of lower bounds for the super-critical regime.

4. On the possible existence of other phase transitions in the average case behavior:

We think that there should be only one phase transition because, to our knowledge, all results on specific graphs exhibit only one transition (note that the existence of a giant component of size a * N implies an average-case influence of at least a^2 * N and therefore all results on sizes of giant components can be used here). However, because our results are upper bounds, they only ensure that a phase transition will occur after \rho_c = 1.

5. Precisions about numerical experiments:

a) In our experiments, except for totally connected networks, edge probabilities are set to the same value p for each edge. When p varies from 0 to 1, the spectral radius of the Hazard matrix varies from 0 to +infinity. We thus used the edge probability as a parameter to tune the spectral radius. b) all points of the plots are averages over 100 simulations. We will include this information in the article. For totally connected networks, we did not display all the simulation points, as the purpose of these very particular networks was to show that the bound was tight. We thus restricted to simulation points close to the upper bound, in order to leave space for other network types.

6. Explanation about equation (13):

There is indeed a notation error in the equation. In the right-hand side, \rho_c(Z^T Z - n_0) is actually \rho_c(A) * (Z^T Z - n_0). The equation follows from (12) with u_i = 1_{i \notin A} * Z_i and the following steps:

u^T Z = Z^T Z - n_0

|u|_1 = |Z|_1-n_0

Z^T H u = Z^T H(A) Z since for all i,j, H_ij u_j = H(A)_ij Z_j

Therefore, (12) implies Z^T Z - n_0 <= (|Z|_1-n_0) (1-exp((Z^T H(A) Z / (|Z|_1-n_0))) which is the left-hand side of (13). Now Z^T H(A) Z <= \rho_c(A) Z^T Z yields the right-hand side.

We agree with the reviewer that this step and others in this area of the paper should be more detailed in order to improve clarity.

7. On misleading notation for H and rho:

We agree that notations could be improved and we shall introduce some slight changes in the revision in order to avoid misinterpretations. In particular, we will take into account the advice of reviewer 6 to use \rho_H instead of \rho_c “to remind the reader that the argument is "pushed through" H first”, for the sake of clarity.

8. About the experimental results for preferential attachment networks:

Our understanding is that our upper bound in Theorem 1 is not highly dependent on the set of influencer nodes, and is rather closer to the maximum influence in the considered network. Thus, inhomogeneous networks with a heavy tailed degree distribution will contain many nodes for which the upper bound will be far from their influence (since peripheral nodes will have a very low influence compared to central nodes). However, some simulation points for preferential attachment networks are close to our bound, which may indicate that the bound is tight for the maximum influence in a network.